# Systematic Review of Computer Vision Semantic Analysis in Socially Assistive Robotics

**Antonio Victor Alencar Lundgren** , **Matheus Albert Oliveira dos Santos** , **Byron Leite Dantas Bezerra** and **Carmelo José Albanez Bastos-Filho \***

Polytechnic School of Pernambuco (POLI), University of Pernambuco (UPE), Recife 50720-001, Brazil; aval@ecomp.poli.br (A.V.A.L.); maos@ecomp.poli.br (M.A.O.d.S.); byron.leite@upe.br (B.L.D.B.)
\* Correspondence: carmelofilho@ieee.org

**Abstract:** The simultaneous surges in the research on socially assistive robotics and that on computer vision can be seen as a result of the shifting and increasing necessities of our global population, especially towards social care with the expanding population in need of socially assistive robotics. The merging of these fields creates demand for more complex and autonomous solutions, often struggling with the lack of contextual understanding of tasks that semantic analysis can provide and hardware limitations. Solving those issues can provide more comfortable and safer environments for the individuals in most need. This work aimed to understand the current scope of science in the merging fields of computer vision and semantic analysis in lightweight models for robotic assistance. Therefore, we present a systematic review of visual semantics works concerned with assistive robotics. Furthermore, we discuss the trends and possible research gaps in those fields. We detail our research protocol, present the state of the art and future trends, and answer five pertinent research questions. Out of 459 articles, 22 works matching the defined scope were selected, rated in 8 quality criteria relevant to our search, and discussed in depth. Our results point to an emerging field of research with challenging gaps to be explored by the academic community. Data on database study collection, year of publishing, and the discussion of methods and datasets are displayed. We observe that the current methods regarding visual semantic analysis show two main trends. At first, there is an abstraction of contextual data to enable an automated understanding of tasks. We also observed a clearer formalization of model compaction metrics.

**Keywords:** lightweight models; computer vision; semantic analysis; socially assistive robotics

## 1. Introduction

With the increase in computing capacity and connectivity, refinement of machine learning algorithms, and aging of the population, among other aspects of modern society, it is expected that robots are to become part of humans' everyday routine in every dimension. Thus, socially assistive robotics (SAR), a field in which robots aid humans in varying social-related tasks, is a significant field of research for our future.

Recent advances in technology, including information technologies, accelerated and diversified SAR studies, including a simulation of companionship [1], assistance with mental health issues [2], and robotic caretakers [3–6]. Those applications create new markets for technology industries aimed at medicine and therapy and expand commodity and easiness of human life.

SAR robots employ sensors and machine learning techniques to accurately perform highly specialized designated tasks. In a large portion of those SAR applications, computer vision (CV) takes an important role, in which robotic platforms coupled with cameras manage to locate and identify humans [7], objects [8], and texts [9].

There are multiple approaches to the CV challenges robots currently face, from the manual feature extractors and set functions to the most recent trend with deep learning models and convolutional neural networks (CNNs). For text and object detection and

recognition, for instance, CNNs show extreme accuracy. Nonetheless, CV challenges concerning SAR persist, mainly mobility, considering the hardware limitations that most mobile robotic platforms present [10].

There are current methods to the mobility problem in research, such as in the works of [11–14], in what defines the research field of lightweight models. Lightweight models can be defined as models that present a low requirement for hardware capabilities, displayed in specific metrics such as FLOPs, number of parameters, storage size, and memory usage.

Although state-of-the-art solves the detection and classification problems in CV robotics platforms relatively well, there remains a vital characteristic for genuine autonomy in assistive robotics: the capability of peripheral analysis of the task at hand. The competence for cognitive jumps to use contextual information favors a solution.

Semantic analysis (SA), defined as the field of machine learning for the mentioned capacity, whether the context surrounding learning artifacts, its most common practices are in natural language processing (NLP), as seen in [15–17], relating to text mining. SA and NLP mainly benefited from the advances in deep learning in general. The critical part of deep architectures realized in the field, including CNNs, was explained in [18].

Referring to and trying to work with constraints on semantics, or anything contextual, can be troublesome since any concept can be linked to context. Thus, throughout this work, we define semantics as the use of indirect information not directly associated with the expected output data, to a goal in terms of machine learning.

Therefore, combining the specified areas of interest can enable hybrid machine learning models to observe an environment, such as a house or room that an older adult or child lives in, and analyze its surroundings in a complex manner, retrieving semantic information to complete the necessary tasks.

Finally, we arrive at a concrete definition for computer vision semantic analysis, which is the use of the secondary information correlated to the primary visual inputs directly extracted from the visual inputs or obtained separately as complementary input and not directly associated to the expected outputs to achieve a given task of computer vision.

For example, one could think of retrieving the position information of an object relative to a room to identify how dangerous the item is, which may also include personal data, to demonstrate that risk may differ from a child to an adult, and from ball bearings on the floor to a knife on the kitchen counter. Another example is retrieving objects' contextual affordance properties, such as shape and size, to identify possible previously unclassified uses.

This step towards VSA can accelerate the emergence of robotic assistants in domestic elder care, allowing for domestic burdens to be removed from the daily tasks of individuals who might not be capable of keeping up with those. Moreover, such robotic agents would guarantee a safer environment for special needs dependent users by repositioning missing, potentially dangerous items, locating lost items, or other relevant but straightforward tasks that would benefit from contextual information.

Given the necessity of merging the knowledge fields for the fully autonomous SAR our future society will require, this work presents a systematic review on socially assistive robots, computer vision, mobile models, and semantic analysis. We aim to study the progress achieved from 2015 to 2020 on this topic and discuss possible future trends based on the knowledge obtained from the state of art. This period was selected due to the more significant advances in machine learning during this period, which is most interesting to our objectives in the context of computer vision and robotics.

We divided this work into the following sections: Section 2 outlines the methodology deployed for the systematic review; Section 3 displays the results of the review; Section 4 answers the research questions and discusses the state-of-the-art; Section 5 discusses the conclusions of this paper.

## 2. Mapping Protocol

Using the guidelines and protocols proposed in the works of [19,20], and therefore following the PRISMA guidelines, we developed our method for planning and realizing our study.

The objective was to collect a corpus of scientific papers in the research field that accounted for specific boundaries and restrictions that make the review useful for authors whatever their final goal and was reproducible by the academic community. For this objective, extensive documentation of the process is necessary.

Consequently, the scope of articles fitting the restrictions is selected and thoroughly analyzed and scored against the relevant research questions defined by the authors before the review.

A significant amount of software have been developed to track papers' status and generate statistics to help authors through the multiple steps of the review. In our work, we used the online software Parsifal https://parsif.al/.

The first step in a systematic review is the definition of its objective (Section 2.1) followed by the research question (Section 2.2), from which the research scope can be shaped. Then, a search strategy is built, including selection criteria, search keywords, and quality assessments.

### 2.1. Research Objective

The main objective of the systematic review proposed in this work was to develop end-to-end models for semantic analysis in computer vision, focusing on their application in assistive robotics. It was idealized to extract context from objects and texts in natural scene images relevant to solving day-to-day problems.

As mentioned in Section 1, we restricted the semantic analysis to the use of surrounding information not directly related to the model's output. Given that simply using the keyword "semantic" results in a high volume of semantic segmentation works, the most broadly observed impact of this definition is the *exclusion of works of semantic segmentation not including visual semantic analysis approaches*.

Semantic segmentation comprises object/room pixel-wise detection and classification. It is important to note that the works of semantic segmentation may still fit into our definition, and those were read and accepted, but semantic segmentation by itself does not.

### 2.2. Research Questions and Search Strategy

We defined essential research questions (RQs) to identify the state of the art concerning semantic analysis in computer vision and its applications and gaps in assistive robotics. We formulated questions to guide the review's directions, define the research fields, and focus the articles' reading and discussion.

Five questions, shown in Table 1, were prepared regarding the connections among vision techniques, robotics, contextual information, and method mobility. To discuss the gaps and trends of visual semantic analysis and its application in assistive robotics, we first need to understand the scope and state of such fields searching for what research and methods currently exist in academia. With that, we defined RQs 1 and 2.

Given that works could be found, we wondered whether there are patterns of semantic data used in works combining computer vision and semantic analysis in the assistive robotics context: is there semantic information that shows itself as most valuable? Are there already defined categorizations of methods for such tasks? This questioning was designed for RQ3.

Our third concern is concerned with the state of the art of lightweight visual semantic analysis models. We find it relevant to gather knowledge on whether this is a current concern of academia. In addition, we would like to analyze what metrics and techniques are used to compare and allow assistant robots to execute those visual tasks. Understanding that the current state and metrics of lightweight models in visual semantics for robotic assistance is one point-of-view and the search for methods for creating lightweight methods is another, we defined RQs 4 and 5.

**Table 1.** Definitions of the research questions that helped mold the research and the points that we planned to study in relation to the state of the art.

| ID | Question | Motivation |
|---|---|---|
| RQ1 | Are there existent approaches for visual semantic analysis in robotic applications? | This question helps identify the current state of autonomous SAR and its current relation to SA. |
| RQ2 | What is the current state of semantic analysis in computer vision? | For our research, it is important to understand the state-of-the-art in vision semantics. |
| RQ3 | What is the most valuable contextual information for assistive robotics? | This question is formulated to investigate what aspects of semantic information are used in SAR and what impact that information has. |
| RQ4 | How lightweight are the current computer vision models and semantic analysis models? | This question allows us to investigate the mobility capacity of current CV and SA methods for SAR. |
| RQ5 | How can machine learning techniques be compacted for domestic robotic use? | This question aids in the search for possible ways robotic machine learning methods can be further compacted. |

From these questions, in conjunction with the defined objective, the research fields approached in our review were made clear, our search, as it seems in Figure 1, is surrounded by semantic analysis, lightweight models, assistive robotics, and computer vision, all of which are related to machine learning.

We defined a set of keywords, period, and the selection and exclusion criteria for our search strategy and selection of the primary studies.

*Criteria for the selection of sources:* The created protocol proposes the use of five academic databases (Table 2). We selected these databases due to their coverage and relevance in engineering and technology, totaling approximately 100 million scientific papers. It is also important to note that we only selected direct academic databases. We did not consider crawlers' search engines for academic purposes to guarantee that a search string was replicable and a stable identifier for each resulting work.

The search was conducted with an automated search for each of the engines listed, using the keywords described below in Table 3, generated from each of the research fields mentioned earlier. Based on the found keywords, the following search string was defined:

*(("assistive robotics" OR "robotic" OR "robotics") AND ("computer vision" OR "image") AND ("semantic analysis" OR "contextual analysis" OR "semantic" OR "object affordance") AND ("compact model" OR "efficient" OR "lightweight" OR "light-weight" OR "mobile model" OR "model compaction")).*

We defined the period as that between 2015 and 2020, six years from the start of the review. It is enough time to understand recent changes and trends of interest. Although the historical information on the field acquirable in older articles is doubtlessly relevant, it does not fit the scope adopted in this paper.

**Table 2.** List of selected sources and their Internet addresses—accessed in January 2022.

| Search Engine | Web Address |
|---|---|
| ACM Digital Library | http://portal.acm.org/ |
| IEEE Digital Library | http://ieeexplore.ieee.org/ |
| Scopus | http://www.scopus.com/ |
| Springer Link | http://link.springer.com/ |
| Web of Knowledge | http://www.webofknowledge.com/ |

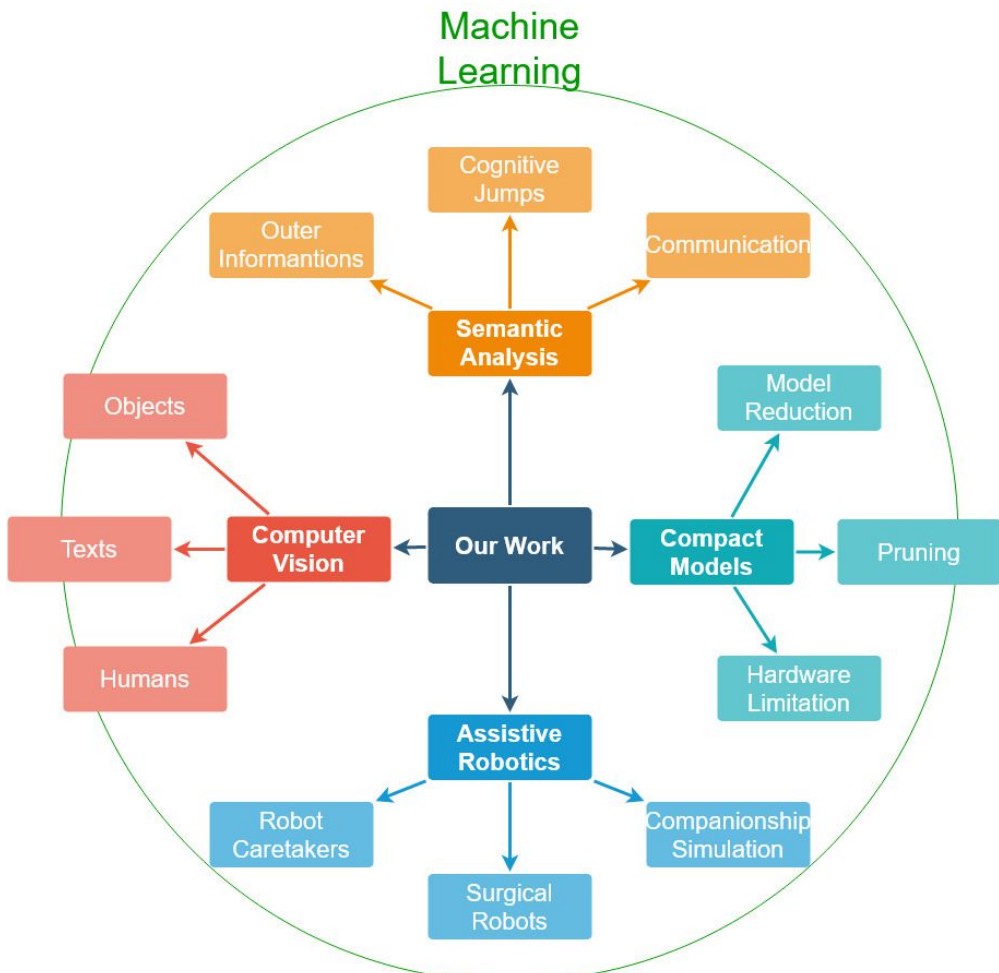

**Figure 1.** Knowledge fields relating to the research objective and formulated research questions.

**Table 3.** Keywords and synonyms used to generate the search string.

| Keyword | Synonymous |
|---|---|
| Assistive robotics analysis | robotic, robotics |
| Compact model | efficient, light weight, light-weight, mobile model, model compression |
| Computer vision | image |
| Semantic analysis | contextual analysis, semantic |

For the inclusion and exclusion criteria, mandatory characteristics for the reviewed papers were defined so that only the works strictly fitted to the scope of our research were kept. For the *inclusion criteria*, the following was defined:

- Inclusion criterion 1: Works including computational methods in computer vision.

For the *exclusion criteria*, the following nine are defined:

- Exclusion criterion 1: Works that are not in English;
- Exclusion criterion 2: Works that are not available in full. A work is considered not available only after contacting the first author and obtaining no response;
- Exclusion criterion 3: Duplicate work;
- Exclusion criterion 4: On-going work;
- Exclusion criterion 5: Works that are a poster, tutorial, editorial, call for papers, chapter, book, or thesis;
- Exclusion criterion 6: Theoretical models that do not currently have a direct implementation;
- Exclusion criterion 7: Works that are literature revisions or surveys;

- Exclusion criterion 8: Works that reached less than five points on the quality criteria;
- Exclusion criterion 9: Works that do not include semantic analysis techniques on images.

*2.3. Research Steps and Information Extraction*

With the research scope delineated, we followed a four-step process for study selection, composed of (1) the study collection; (2) a preliminary selection process; (3) a final selection process; and (4) a quality assessment of the selected studies. We applied the search string to the conventional study sources during the study collection. In this step, the first exclusion that was directly applied to the search engines of each academic database was the definition of the period and text language.

The titles and abstracts of the papers were briefly read through following the preliminary selection process. The texts were divided among the pair of reviewers. Each reviewer selected or excluded from their list of papers, defining at least on exclusion criterion as the reason for exclusion.

The papers selected in the previous step were read in total during the third step to search for false positives. This step was performed in pairs. Each paper was entirely read by both evaluators and given approval or exclusion, again accompanied by the exclusion criterion or criteria. In case of nonconformity in results, the article in question was discussed.

Finally, during the quality assessment, which occurs in conjunction with the third step, the selected papers were scored with the quality questions (Table 4). For each question of the checklist, the following scale was used: yes (Y) = 1 point; no (N) = 0 points; partially (P) = 0.5 points. Furthermore, the papers were critically evaluated and discussed to understand the nuances of the field in general.

**Table 4.** Quality assessment criteria and possible scores.

| ID | Quality Criteria | Answers |
|----|------------------|---------|
| QC1 | Is there a description of the application domain in which the research was conducted? | Yes/No/Partially |
| QC2 | Is the work aimed towards robotics in the home assistance of those with special needs? | Yes/No/Partially |
| QC3 | Is the implementation of the method publicly available? | Yes/No/Partially |
| QC4 | Is there a description of the dimensions/factors considered? | Yes/No/Partially |
| QC5 | Is there a description of the datasets used? | Yes/No/Partially |
| QC6 | Are the datasets used publicly available? | Yes/No |
| QC7 | Are the results are clearly reported? | Yes/No/Partially |
| QC8 | Do the results add value to the area of research? | Yes/No/Partially |

Afterward, aiming for the discussion over the selected papers, the adopted information extraction strategy was that, for each selected study, the following data would be extracted:

1. Title of the article;
2. Name of the authors;
3. Year;
4. Search engine/base;
5. Model type;
6. Application domain;
7. Datasets;
8. Requirements of the model;
9. Result metrics.

### 3. Results

At the end of the first step of the study selection (see Section 2.3, the initial research corpus was composed of 670 papers, of which 330 were acquired from the ACM Digital Library; 124 from SpringerLink; 111 from IEEE Digital Library; 81 from Scopus; and 24 from Web of Knowledge, as seen in Figure 2.

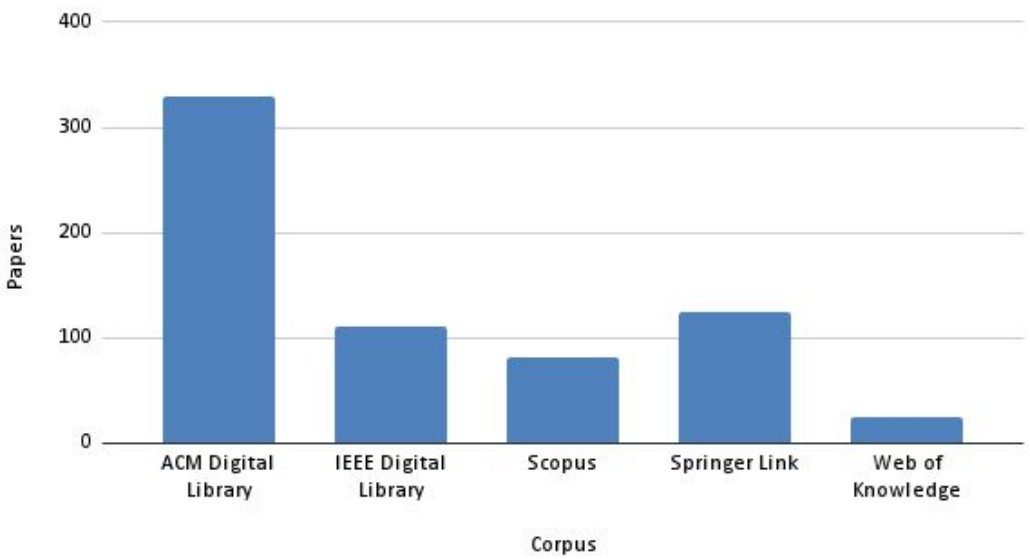

**Figure 2.** Initial corpus division by scientific articles database.

It is, however, important to reinforce that the numbers obtained in step 1 were obtained previously to any exclusion due to analysis and contain repetitions among different databases. Subsequently, the studies were divided at the start of the second step. This reading is composed of titles and abstracts searching for any information that may put the article in one of the nine exclusion criteria. Of the initial 670 studies, only 91 were selected through this first reading.

During this review phase, the most common exclusion reason was criterion 9 (works that do not include semantic analysis techniques on images), with 376 works excluded. A large majority of those papers were included due to the keywords amassing many semantic segmentation works without using semantic analysis. Criterion 9 was also most commonly paired with criterion 6 (theoretical models that do not currently have a direct implementation), with 57% of concurrences of the latter with criterion 9.

Further data on the remaining exclusion criterion in step 2 are shown in Figure 3.

The third step of the systematic review had the remaining 91 papers read in full. This secondary reading, as explained in Section 2.3, aimed to identify possible unnoticed false positives. This step was demanding, with the reading performed in pairs, and both reviewers needing to agree on each study outcome.

The entire reading showed that many of the works previously believed to relate to computer vision were not related to the abstract reading. Moreover, many works mistaken as semantic analysis works were still studies of semantic segmentation. Among the 91 previously selected, 57 were again excluded by exclusion criterion 9 alone, and 1 was the same work in which each reviewer read one copy. The final count was 33 papers accepted for quality assessment.

The last step of the reviewing process was developed simultaneously with the previous one. When one work was fully read, if it was not excluded, the reviewers wrote notes about the work and graded the paper against the quality criteria introduced in Section 2.3.

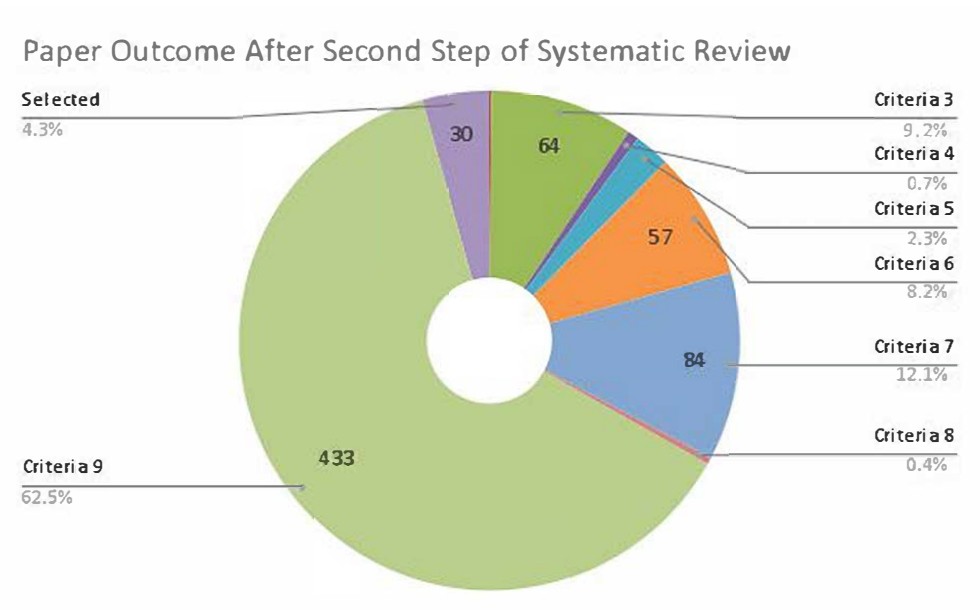

**Figure 3.** Division in the study outcome in selected or excluded works after the end of the second step. Excluded works are shown by exclusion criteria. Works may have more than one criterion associated with the exclusion. There are occurrences in the exclusion criteria.

During this phase, exclusion criterion 8 was applied. The three last works were excluded for not obtaining the minimum score, totaling 30 accepted papers in this systematic review. Figure 4 illustrates the pipeline that the collected works went through.

PRISMA 2020 flow diagram for new systematic reviews which included searches of databases and registers only

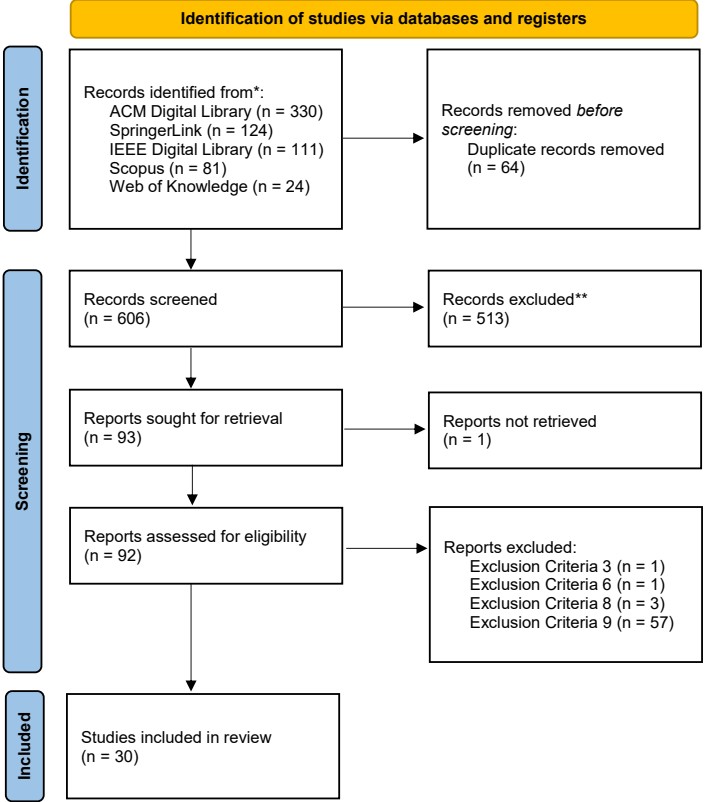

**Figure 4.** The systematic review's pipeline and number of papers selected at each step, shown as the PRISMA flow diagram.

We observed the following distribution regarding the scientific papers databases: 11 from Springer Link; 11 from ACM Digital Library; 4 from IEEE Digital Library; and 2 from both Scopus and Web of Knowledge.

Another important piece of data collected from the accepted papers during the reviews concerned the publishing year: 30.00% of the accepted papers are from 2020; 30.00% are from 2019; 10.00% are from 2018; 10.00% are from 2017; 13.33% are from 2016; and 6.67% are from 2015.

Each of the works' scores in the quality assessment is displayed in Table 5 in a decreasing order.

**Table 5.** Attributed quality scores for each of the approved works in decreasing order.

| Paper | Authors | Score |
|---|---|---|
| Importance-Aware Semantic Segmentation with Efficient Pyramidal Context Network for Navigational Assistant Systems | [21] | 7.0 |
| O-PrO: An Ontology for Object Affordance Reasoning | [22] | 7.0 |
| Enhancing V-SLAM keyframe selection with an efficient convnet for semantic analysis | [23] | 6.5 |
| Finding misplaced items using a mobile robot in a smart home environment | [24] | 6.5 |
| Hierarchical Character Embeddings: Learning Phonological and Semantic Representations in Languages of Logographic Origin Using Recursive Neural Networks | [25] | 6.5 |
| LGNN: A Context-Aware Line Segment Detector | [26] | 6.5 |
| Behavioural Pattern Discovery from Collections of Egocentric Photo-Streams | [27] | 6.0 |
| Boosting Real-Time Driving Scene Parsing With Shared Semantics | [28] | 6.0 |
| Multimedia Knowledge Design and Processing for Personal Robots | [29] | 6.0 |
| Understanding of Human Behavior with a Robotic Agent Through Daily Activity Analysis | [30] | 6.0 |
| Attribute Based Affordance Detection from Human-Object Interaction Images | [31] | 5.5 |
| Interaction Context (ICON): Towards a Geometric Functionality Descriptor | [32] | 5.5 |
| Scene Recognition with Sequential Object Context | [33] | 5.5 |
| Substitute Selection for a Missing Tool Using Robot-Centric Conceptual Knowledge of Objects | [34] | 5.5 |
| Added Value of Gaze-Exploiting Semantic Representation to Allow Robots Inferring Human Behaviors | [35] | 5.0 |
| An approach to provide dynamic, illustrative, video-based guidance within a goal-driven smart home | [36] | 5.0 |
| Attention-Based Generative Graph Convolutional Network for Skeleton-Based Human Action Recognition | [37] | 5.0 |
| Fusion of range measurements and semantic estimates in an evidential framework | [38] | 5.0 |
| Grounding Visual Concepts for Zero-Shot Event Detection and Event Captioning | [39] | 5.0 |
| Knowledge Acquisition and Design Using Semantics and Perception: A Case Study for Autonomous Robots | [40] | 5.0 |
| Scene-Dependent Intention Recognition for Task Communication with Reduced Human-Robot Interaction | [41] | 5.0 |
| Semantic event fusion of computer vision and ambient sensor data for activity recognition to support dementia care | [42] | 5.0 |
| Where Should I Walk? Predicting Terrain Properties From Images Via Self-Supervised Learning | [43] | 5.0 |
| End-to-End Training of Deep Visuomotor Policies | [44] | 4.5 |
| LifeBots I: Building the Software Infrastructure for Supporting Lifelong Technologies | [45] | 4.5 |
| A semantic-based gas source localization with a mobile robot combining vision and chemical sensing | [46] | 4.0 |

**Table 5.** *Cont.*

| Paper | Authors | Score |
|---|---|---|
| Mini: A New Social Robot for the Elderly | [47] | 4.0 |
| The head turning modulation system: An active multimodal paradigm for intrinsically motivated exploration of unknown environments | [48] | 4.0 |
| Deep Learning Based Glance of Real World Scenes through Decision Tree | [49] | 3.5 |
| Multimodal Fuzzy Assessment for Robot Behavioral Adaptation in Educational Children-Robot Interaction | [50] | 3.5 |

### 3.1. Tasks and Datasets

The works resulting from the review present a large variety of tasks in CV, from object detection to emotion classification. This corresponds to our expectations since, as discussed before, contextual information is not strictly attached to a single objective.

Here, we discuss the vision tasks observed in the papers in a general manner. For most accepted research, the CV tasks are the main goal or the primary data to achieve the desired goal. However, those tasks are not always the end goal of the work. For some [22,32,44,46], the visual information itself is used as secondary information or a means to obtain these mid-to-high level data judged necessary to an objective.

We cluster the tasks into broad areas: area (1)—object recognition, (a) including object detection, (b) object classification, and (c) object part recognition; area (2)—human recognition, composed of (a) emotion classification, (b) human detection, (c) gaze detection, (d) pose classification, (e) activity classification, (f) and goal prediction; area (3)—text recognition; and area (4)—scene recognition, divided into (a) semantic segmentation, and (b) scene classification.

As shown in Table 6, the most common tasks are related to object recognition followed by human-related recognition. We found the datasets applied in the papers to be as varied as the CV research areas. Again, here we only discuss vision-related datasets.

**Table 6.** Computer vision tasks approached for each accepted paper, clustered into four broad groups.

| No. | Ref. | Objects | Humans | Text | Scenes |
|---|---|---|---|---|---|
| 1 | [46] | ✓ | | | |
| 2 | [35] | ✓ | ✓ | | |
| 3 | [36] | | ✓ | | |
| 4 | [37] | | ✓ | | |
| 5 | [31] | ✓ | ✓ | | |
| 6 | [27] | ✓ | ✓ | | ✓ |
| 7 | [28] | | | | ✓ |
| 8 | [49] | | | | ✓ |
| 9 | [44] | ✓ | | | |
| 10 | [23] | | | ✓ | ✓ |
| 11 | [24] | ✓ | | | |
| 12 | [38] | | | | ✓ |
| 13 | [39] | | | ✓ | |
| 14 | [25] | ✓ | | | |
| 15 | [21] | | | | ✓ |
| 16 | [32] | ✓ | | | |
| 17 | [40] | ✓ | | | |
| 18 | [26] | | | | |
| 19 | [45] | ✓ | | | ✓ |
| 20 | [47] | ✓ | ✓ | | |
| 21 | [29] | ✓ | | | |
| 22 | [50] | | ✓ | | |

**Table 6.** *Cont.*

| No. | Ref. | Objects | Humans | Text | Scenes |
|-----|------|---------|--------|------|--------|
| 23 | [22] | ✓ | | | |
| 24 | [33] | | ✓ | | |
| 25 | [41] | ✓ | | | ✓ |
| 26 | [42] | | ✓ | | |
| 27 | [34] | ✓ | | | |
| 28 | [48] | ✓ | ✓ | | |
| 29 | [30] | ✓ | ✓ | | |
| 30 | [43] | | | | ✓ |

For object recognition tasks, four datasets were identified. The large-scale natural scene object detection and classification dataset *COCO* https://cocodataset.org/, was in [29,49]. The *Imagenet* https://www.image-net.org/ dataset with millions of images following the WordNet hierarchy was in [22,29,41]. *Wireframe* https://github.com/huangkuns/wireframe (accessed on 30 June 2021), the object-to-line regression dataset, was in [26]. *RGB-D (Kinect) Object* http://rgbd-dataset.cs.washington.edu/ (accessed on 30 June 2021) is an RGB-D dataset for 3D object recognition identified in [34].

The authors of [45] applied *ViDRILO* rovit.ua.es/dataset/vidrilo/ (accessed on 30 June 2021), a scene classification dataset which also contains annotations for objects in scene. Likewise, for scene classification problems, Authors Menchon et al. [27] worked on the *Places365* http://places2.csail.mit.edu/download.html (accessed on 30 June 2021) dataset, and [51] combined both the *Places* and *Sun* https://vision.princeton.edu/projects/2010/SUN/ (accessed on 30 June 2021) scene classification dataset in their work.

For the tasks of action recognition, four datasets were encountered. *CMU-MMAC* http://kitchen.cs.cmu.edu/, found in [35], contains information from video, audio, motion capture, and other sensors for classifying cooking activities. The *EgoRoutine* http://www.ub.edu/cvub/dataset/egoroutine/ (accessed on 30 June 2021) dataset presents egocentric photo streams annotated towards behavior characterization applied in [27]. The *NTU-RGB+D* https://rose1.ntu.edu.sg/dataset/actionRecognition/ (accessed on 30 June 2021) contains video, infrared, and depth information about multiple actions. Finally, the *Kinetics* https://git-disl.github.io/GTDLBench/datasets/kinetics_datasets/ (accessed on 30 June 2021) dataset was identified, which is composed of depth information for activities. The latter two are combined in [37].

For problems of text recognition, the model proposed by [23] was trained on the *COCO-Text* https://vision.cornell.edu/se3/coco-text-2/ (accessed on 30 June 2021) dataset, a re-purposing of the COCO dataset in the context of natural scene text detection and recognition.Authors Nguyen et al. [25] trained on the *Unihan* http://www.unicode.org/charts/unihan.html (accessed on 30 June 2021) dataset for logographic characters.

Four more known public datasets were identified in the papers regarding semantic segmentation, *Cityscapes* https://www.cityscapes-dataset.com/dataset-overview/ (accessed on 30 June 2021) in [21,23,38] and *CamVid* http://mi.eng.cam.ac.uk/research/projects/VideoRec/CamVid/ (accessed on 30 June 2021) as well as in [21] model outdoor scene segmentation, whilst *MIT 67* http://web.mit.edu/torralba/www/indoor.html (accessed on 30 June 2021) is used by [41] for indoor segmentation purposes. Table 7 displays all the datasets used in the references.

**Table 7.** References presented by the used datasets. Private or non-specified datasets are not presented herein.

| Task | Dataset | References |
|------|---------|-----------|
| Semantic segmentation | Cityscapes | [21,23,38] |
| | CamVid | [21] |
| | MIT 67 | [41] |

**Table 7.** *Cont.*

| Task | Dataset | References |
|------|---------|-----------|
| Scene classification | Places365<br>Places<br>Sun<br>ViDRILO | [27]<br>[41,49]<br>[41,49]<br>[45] |
| Object recognition | COCO<br>Imagenet<br>Wireframe<br>Washington Dataset | [49]<br>[22,29,41]<br>[26]<br>[34] |
| Action recognition | CMU-MMAC<br>NTU-RGB+D<br>EgoRoutine<br>Kinetics | [35]<br>[37]<br>[27]<br>[37] |
| Text recognition | COCO-Text<br>UniHan | [23]<br>[25] |

The work in [35] provides two novel datasets for cooking-related action recognition; Authors Rafferty et al. [36] published in their work a dataset for video description; Xiang et al. [28] contribute an outdoor semantic segmentation dataset focused on automatic driving vehicle; and Hu et al. [32] and Bhattacharyya et al. [22] bring object affordance ontologies to be used as semantic data in various applications.

Furthermore, the works [24,31,33,39,40,42,44,46–48,50] either do not specify the data used for training the ML models displayed or use private datasets.

### 3.2. Machine Learning Methods

As can be observed in Table 8, with regard to the machine learning methods for CV, 63.33% of the works use deep learning techniques. However, many other approaches are taken from probabilistic machine learning (PML) to decision trees (DT) and unsupervised machine learning (UML).

**Table 8.** The machine learning techniques utilized in each work. Unk. means that the papers do not specify what method they used. GNNs are graph neural networks. CNNs are convolutional neural networks. RNNs are recurrent neural networks. UML stands for unsupervised machine learning. SVMs are support vector machines. PML stands for probabilistic machine learning. Finally, DTs are decision trees.

| No. | Ref. | Unk | GNNs | CNNs | RNNs | UML | SVMs | PML | DTs |
|-----|------|-----|------|------|------|-----|------|-----|-----|
| 1 | [46] | | | ✓ | | | | ✓ | |
| 2 | [35] | | | | | | | | ✓ |
| 3 | [36] | | | | | | | | |
| 4 | [37] | | ✓ | ✓ | | | | | |
| 5 | [31] | | | | | ✓ | ✓ | ✓ | |
| 6 | [27] | | | ✓ | | | | | |
| 7 | [28] | | | ✓ | | | | | |
| 8 | [49] | | | ✓ | | | | | ✓ |
| 9 | [44] | | | ✓ | | | | | |
| 10 | [23] | | | ✓ | | | | | |
| 11 | [24] | | | ✓ | | | | ✓ | |
| 12 | [38] | | | ✓ | | | | | |
| 13 | [39] | | | | ✓ | | ✓ | | |
| 14 | [25] | | | | | | | | |
| 15 | [21] | | | ✓ | | | | | |
| 16 | [32] | | | | | | ✓ | | |

**Table 8.** *Cont.*

| No. | Ref. | Unk | GNN | CNN | RNN | UML | SVM | PML | DT |
|-----|------|-----|-----|-----|-----|-----|-----|-----|-----|
| 17 | [40] | | | ✓ | | | | | |
| 18 | [26] | | ✓ | ✓ | | | | | |
| 19 | [45] | | | ✓ | | | | | |
| 20 | [47] | ✓ | | | | | | | |
| 21 | [29] | | | ✓ | | | | | |
| 22 | [50] | | | ✓ | | | | | |
| 23 | [22] | | | | | | | | |
| 24 | [33] | | | | | | ✓ | ✓ | |
| 25 | [41] | | | ✓ | | | | | |
| 26 | [42] | | | | | | ✓ | | |
| 27 | [34] | | | | | | ✓ | | |
| 28 | [48] | ✓ | | | | | | | |
| 29 | [30] | | | | | | | | ✓ |
| 30 | [43] | | | ✓ | | | | | |

### 3.2.1. Deep Learning Methods

The most commonly employed methods are various convolutional neural networks (CNN) architectures that follow the current global trend in CV. A few of the works reuse pre-trained already established CNN models, such as [52,53].

Authors yang et al. [37] propose a novel AG-GCN, an attention-based graph convolutional network that combines CNN with GNN to achieve state-of-the-art results with 94.3% accuracy on cross-view and 86.1% cross-subject accuracy, both on the NTU dataset for action recognition. A novel CNN model for visuomotor policy prediction is presented in [44], trained on a private dataset, achieving up to 100% of accuracy.

An average precision of 62.3% in the Wireframe dataset is achieved in [26] by introducing the line graph neural network (LGNN), a three-branch network using CNN as a feature-extractor for a graph neural network (GNN).

An extremely lightweight CNN architecture for semantic segmentation, optimized towards CPU use, was proposed by [23], in which accuracy is traded for mobility, achieving 36.69% precision on COCO-Text and 70.5% category IoU on the CityScapes dataset. BiERF-PSPNe for semantic segmentation was presented in [21], obtaining a mean precision of 76.3% on the Cityscapes dataset.

A terrain classification CNN was applied and the self-supervised model was proposed by [43] for robot path-planning, in which the data provided are private and the model achieves up to 97.3% accuracy.

In the work of [28], a two-branched fully convolutional network (FCN) was presented. One branch is an unspecified lightweight CNN and the other is a semantic segmentation [54]. Both branches are fused. The work obtained state-of-the-art results of up to 99.6% accuracy on the CAM-120 dataset.

### 3.2.2. Non-Deep Learning Methods

In [39], a hierarchical recurrent neural encoder (HRNE) [55] was used for multimedia event captioning on a private unspecified dataset.

K-nearest neighbors (KNNs) algorithms were used in [32] to search for object similarity. Furthermore, Hassan and Dharmaratne [31] applied a combination of KNNs, support vector machines (SVMs), and probabilistic machine learning in the form of Bayesian networks as classifiers for object and human affordance characteristics, such as color, orientation, and body pose.

RANSAC-based algorithms were presented in [33] with RGB-D point cloud segmentation and with SVM as object classifier. Similarly, Stavropoulos et al. [42] used SVM for activity classification, achieving up to 82% precision in the unspecified dataset.

A Bayesian network was demonstrated in [56], merging inputs from nonvisual sensors with object affordance characteristics obtained after object classification with CNN to

calculate the certainty of the origin of the smell. Authors Kostavelis et al. [30] trained a dynamic Bayesian network for activity classification on a video on a private dataset, for human–robot interaction, attaining an average precision and recall of over 98%.

The C4.5 decision tree (DT) algorithm was exploited in [35] for human activity classification on the author's public dataset, obtaining precision rates of up to 80%. Likewise, Pawar et al. [49] trained a DT algorithm for scene classification merging with CNN, achieving 97% accuracy.

*3.3. Computer Vision Semantic Analysis*

In regard to computer vision semantic analysis, we defined three main methods for using contextual information which were identified in the accepted works:

- **Ontology**, in which the work proposes or utilizes a knowledge base based on experts and built by grouping semantic contexts in a way to classify the base concepts deemed important for a goal;
- **Information merging**, through multiple inputs or sensors, or grouping of outputs as re-purposed input, this method exploits surrounding information which can be used for the final goal;
- **Semantic data extraction**, by algorithms, usually ML models outputs, high-level data are used as input for other means. It uses usually considered nonsemantic knowledge obtained by such algorithms and transforms it into contextual information pertinent to the final goal. This method is usually linked to information merging.

Table 9 presents the application of the three main visual semantic analysis methods throughout the accepted papers.

**Table 9.** Identified method for visual semantic analysis.

| No. | Ref. | Ontology | Info. Merging | Semantic Data Extraction |
|-----|------|----------|---------------|--------------------------|
| 1 | [46] | ✓ | ✓ | |
| 2 | [35] | ✓ | ✓ | |
| 3 | [36] | ✓ | | |
| 4 | [37] | | ✓ | ✓ |
| 5 | [31] | | | ✓ |
| 6 | [27] | | ✓ | |
| 7 | [28] | | | |
| 8 | [49] | | ✓ | ✓ |
| 9 | [44] | | ✓ | |
| 10 | [23] | | | ✓ |
| 11 | [24] | | ✓ | |
| 12 | [38] | | ✓ | |
| 13 | [39] | | ✓ | ✓ |
| 14 | [25] | | | ✓ |
| 15 | [21] | | | ✓ |
| 16 | [32] | ✓ | | |
| 17 | [40] | | | ✓ |
| 18 | [26] | | | ✓ |
| 19 | [45] | | ✓ | |
| 20 | [47] | | ✓ | |
| 21 | [29] | ✓ | ✓ | |
| 22 | [50] | | ✓ | |
| 23 | [22] | ✓ | | |
| 24 | [33] | | | ✓ |
| 25 | [41] | | ✓ | ✓ |
| 26 | [42] | ✓ | ✓ | |
| 27 | [34] | | ✓ | ✓ |
| 28 | [48] | | ✓ | |
| 29 | [30] | | ✓ | ✓ |
| 30 | [43] | | ✓ | |

### 3.3.1. Ontologies

As for ontologies, eight papers used this method of semantic analysis. Monroy et al. [46] presented an object affordance ontology in the context of object smell, modeling a "thing" as either inert or living, including features such as size, orientation, and smell. Ramirez et al. [35] modeled an object-use ontology for robotic encounters, in which the characteristics of the actions connecting humans and objects are modeled.

An ontology for the task of activity goal was used by [36], in which steps and states necessary to act are defined. A framework for multimedia ontologies using WordNet and ImageNet was presented by [29], in which the focus is human–robot and device–robot interactions and the ontology defines organisms and objects in multiple levels of hierarchy.

O-Pro, an ontology for object reasoning, was introduced in [22] including general object characteristics such as shape, weight, volume, and concavity, which are modeled to correspond to object classes. Stavropoulos et al. [42] displayed a knowledge base for human–object interaction as a way to classify activities.

### 3.3.2. Information Merging

Regarding information merging, two main groups were identified: (1) those who exploited multiple sensors, those being cameras or nonvisual sensors; and (2) those who merged multiple machine learning models' outputs as input for a final model.

As examples of the former group, we have: Monroy et al. [46], using camera-based information and electronic-nose sensors as combined input for smell detection; Xiang et al. [28] for utilizing two cameras of different intersecting views as input to an FCN model; and [42], which presented a framework for merging camera data with a multitude of different sensors in elder care systems.

Examples for the latter group include [41], which took the output of an object classification CNN and merged the data into a scene recognition model, and [49], where the outputs of a CNN for object detection were applied in a decision tree method to classify a scene. An SVM trained to detect object concepts in video frames was used together with an RNN model for video description in [39].

### 3.3.3. Semantics Data Extraction

For semantic data extraction, we mainly observed detection or classification ML models as input or combined input for an ML model for the paper goal. As an example, Wang et al. [41] detect objects in a scene and uses this detection as contextual information, together with the entire scene image for scene classification.

In [34], an example for semantic data extraction before execution is provided; CNN object detections are clustered together to provide tool substitution predictions. Metasemantic data were derived in [23], where video frames have their contextual relevance predicted by an ML model, in which only the most relevant frames are processed, thus reducing the method complexity.

## 4. Discussion

The work found in this systematic review satisfied our search in their variety of tasks, methods, and uses for semantic information, allowing us to identify multiple research gaps.

Most importantly, we believe that there is a clear path towards computer vision semantic analysis by direct extraction of contextual information from pixel data, not observed in any of the works. Instead of building knowledge from the top down, as ontologies unavoidably are, it might be possible to let modern techniques understand the more granular information in those knowledge bases directly from images.

Now, we could remove the need for an object to be tagged for its affordance characteristics to be derived, but the inverse, predicting affordance characteristics, would then be utilized for achieved whatever goal is desired.

Furthermore, some of the works used older methods; even some which used CNN approaches used these methods as midways for another, such as probabilistic networks [24,46], decision trees [49], or SVMs [39]. This means these could most likely be improved.

The rest of this section discusses the specific topics relevant to the authors and answers the raised research questions.

*4.1. RQ1: What Is the Current State of Semantic Analysis in Computer Vision?*

We observed that associations of SA and CV are in their starting steps through the gathered information. Although we can see semantic analysis applications with the creation of visual ontologies and merging external information to improve accuracy in visual tasks, the definition of what contextual data are essential seems to be a recurring challenge.

We can mostly find the merging of ML model outputs, which, when applied in particular contexts, seems to improve the model accuracy. As it is found in its starting steps, many challenges still permeate the field: how can contextual information be transferred, akin to what is currently done in transfer learning? Can mid-level contextual information be extracted, adapted, and more generally utilized for diverse tasks similar to the processes humans make? Could that mid-level contextual information be directly extracted from images without the need for unmodifiable expert-given knowledge, such as ontologies?

Currently, the main difficulty is to model the cognitive data jumps extracted from general information in images/videos. Moreover, most works aimed to either add nonvisual information as context or attach visual data to graph knowledge bases to acquire the necessary push towards SA.

Furthermore, the increase in interest in the field, presented in Section 3, presents a rapidly growing trend, with the number of published papers more than doubling during the period included in this review. As techniques from fields more commonly related to AS, such as natural language processing, are brought to CV, we predict more efficient approaches will emerge in this area.

*4.2. RQ2: Are There Existent Approaches for Visual Semantic Analysis in Robotic Applications?*

We found that there are visual SA tasks performed by robotics platforms [21,24,28–31,33–35,40,43–48,50], specifically SAR applications, which we were glad to find as those most closely related to our personal goals.

Semantic analysis in those works followed the general trend of information merging and secondary data generation from visual input. Nevertheless, most of the displays of nonvisual sensors as a source of merging data come from those works, with only two exceptions of the seven works using this semantic analysis technique.

As for SAR applications, a focus was found, including elder task relief, such as finding lost objects [24]. Nevertheless, those are a minority of the accepted works.

*4.3. RQ3: What Are the Most Valuable Contextual Information Data for Assistive Robotics?*

As directly observed in Section 3.3, we identified that the most common manner of using semantic information is by information merging, that being by using multiple sources of data (i.e., multiple sensors as input data) or by deriving new data from one single camera input (e.g., merging outputs from ML models of different purposes), with more than half the works applying some variation of this method of visual semantic analysis.

As for one single datum or data category, used as contextual information in ML models for SA, unfortunately, we cannot accurately answer this question with the current work. As expected, contextual information is highly relative to the challenge at hand and wildly varying in each approved study—for example, chemical information used for object detection [46] or pressure data in path planning [43].

In general, despite taken as a classical approach, we can classify object affordance as promising for visual semantics SAR applications. With five accepted papers working around object usability descriptions [22,31,32,34,44], this approach can lead to robots better

understanding what their visual sensors show. In the future, it may also benefit from ML techniques that are not available or have not been previously researched.

Moreover, combining the multiple SA methods appearing in the accepted papers might improve SAR systems.

### 4.4. RQ4: How Lightweight Are the Current Computer Vision Models and Semantic Analysis Models?

To our surprise, mobility metrics are not standard in the accepted works. Some of the works are directly applied to robots, meaning that they can run on the specific platform tested on, and some specify pseudo-metrics for mobility, such as frames per second of model processing.

However, the lack of compactness information, such as FLOPs and RAM usage, makes the described methods not certainly portable to other platforms. In this way, CV models appear to be compacted to run on intended robotic systems; however, we cannot scientifically assure how lightweight those models are.

Moreover, compactness is highly dependent on the model utilized, with lighter CNN architectures available for more limited hardware. As an example, we can see that MobileNet is used in [49], a lightweight CNN with 1B FLOPs and 4.2M parameters. Compared with the architecture on [50], based on the YOLOv3 model, which calculates 65B FLOPs and contains 65M parameters, it is visible how the former is better optimized towards limited robotic platforms.

### 4.5. RQ5: How Can Machine Learning Techniques Be Compacted for Domestic Robotic Use?

Attempting to compact models is vital for methods such as deep learning which tend to increase computational execution requirements. One work, Alonso et al. [23], the only one to explicitly contribute in compacting techniques, tried to make the process less burdening on hardware by creating a method for key-frame selection and combining it with a lightweight CNN model that rapidly reduces the feature map resolution, resulting in less convolutional operations.

This RQ was set to understand how papers in intersecting target fields exposed in Section 1 treat robot hardware limitations when performing consuming tasks, such as CV-focused deep learning. Although this was not the expected outcome, from the mentioned work, we can take that visual semantic information may be helpful in itself in reducing computing costs by contextually deciding which information is most beneficial to the process.

Nonetheless, RQ4 and RQ5 proved a challenge in the current work, as even papers aimed at robotics rarely displayed technical data to model mobility, such as FLOPs, memory usage, and others. Outside of the scope of this review, there are multiple methods for pruning and reducing the size of a deep learning model. As these models are picked up in the visual semantic SAR tasks, we are more likely to see compacting methods applied.

### 4.6. Future Trends

There are two main gaps identifiable in the current state of the art: (1) automatic extraction of semantic information; (2) focus on compactness.

The first gap is the most important. The automatic extraction of relevant semantic information without ontologies should be the significant threshold for CV tasks of SA, which is partially achieved through the examples of information merging discussed before. However, current DL methods for CV can allow for more direct pipelines of semantic data extraction and generalization towards a goal.

More specifically, we believe data for object affordance, currently attached to well-established knowledge in the shape of ontologies, could be directly extracted from pixels similarly to other DL visual tasks, abandoning usual tagging concepts on which affordances are usually built while not, by nature, giving up on the readability of data. Furthermore, those methods could be used as decision-making information for generalized SAR.

Current powerful techniques, although uncertain, are a trend to pursue in the search for a valid autonomous SAR. CV deep learning models are currently used in SAR, exclusively in the already defined tasks of classification, detection, recognition, and segmentation. However, there is space for novel tasks to be approached by such models or new models.

How can be the usability of objects be automatically modeled by machine learning? What kind of semantic data can be extracted by intelligent agents? Can semantic information be directly obtained without human expertise? These are some of the significant challenges for the near future of the field.

For the second gap, in this work's scope, mobility is not commonly explicitly detailed or worried. As more computationally complex tasks appear, field work needs to focus on compacting issues, detailing platform-agnostic metrics. Those metrics must become the norm in SAR work towards scientific replication.

## 5. Conclusions

In this work, we introduced a systematic review on the overlapping research fields of computer vision, semantic analysis, socially assistive robotics, and lightweight models. We presented the following main contributions:

- We provided a concrete definition of computer vision semantic analysis which can identify works in the fields of computer vision and semantic analysis, as described in Section 2;
- We present a systematic review of the current scope, research gaps, and future trends on the field of computer vision semantic analysis and its applications towards assistive robotics;
- We define a novel classification of methods in computer vision semantic analysis: (1) ontologies; (2) information merging; (3) semantic data extraction.

Six hundred and seventy studies were collected in five scientific databases using the keywords relevant to the field. In a four-step exclusion of articles not matching the defined search scope, 30 papers were approved and analyzed in the context of predefined research questions. The accepted papers discussed their role and contributions in the specific area of interest and scored in quality from 3 to 8 points. The maximum score was 7, obtained by two studies: [21,22].

Methods for computer vision ML operations and computer vision semantic analysis are displayed, as are the datasets used for the accepted works. For computer vision semantic analysis, we present a novel categorization, divided into three groups: ontologies, information merging, and semantic data extraction.

Based on the analyzed works, we may also conclude that the use of computer vision semantic analysis can improve accuracy, as seen in [37,43,44,54]. This points towards exploring novel vision tasks and the improvement of current challenges. Finally, this review discusses possible research gaps and future state-of-the-art trends on visual semantic SAR challenges.

Finally, one prospect for future work is associated with the compactness of robotic solutions and the standard of lightweight metrics detailing. Another gap is considered a significant breakthrough in the future of this field: the automatic extraction of contextual information by machine learning.

**Author Contributions:** Conceptualization, A.V.A.L.; methodology, A.V.A.L. and M.A.O.d.S.; writing—original draft preparation, A.V.A.L.; writing—review and editing, A.V.A.L., B.L.D.B. and C.J.A.B.-F.; supervision, B.L.D.B. and C.J.A.B.-F.; funding acquisition, B.L.D.B. and C.J.A.B.-F. All authors have read and agreed to the published version of the manuscript.

**Funding:** This research was financed in part by the Coordenação de Aperfeiçoamento de Pessoal de Nível Superior—Brasil (CAPES)—Finance Code 001, by Fundação de Amparo à Ciência e Tecnologia do Estado de Pernambuco-Brasil (FACEPE), and Conselho Nacional de Desenvolvimento Científico e Tecnológico-Brasil (CNPq).

**Institutional Review Board Statement:** Not applicable.

**Informed Consent Statement:** Not applicable.

**Data Availability Statement:** All publicly available datasets are discussed in Section 3.1 with links found in the footnotes. Furthermore, the accepted papers were extracted from the databases discussed in Section 2: the results are replicable using the search string defined in Section 2.2.

**Conflicts of Interest:** The authors declare no conflict of interest.

## Abbreviations

The following abbreviations are used in this manuscript:

| | |
|---|---|
| CNN | Convolutional Neural Network |
| CV | Computer Vision |
| DT | Decision Tree |
| FCN | Fully Convolutional Network |
| GNN | Graph Neural Network |
| HRNE | Hierarchical Recurrent Neural Encoder |
| KNN | K-Nearest Neighbors |
| LGNN | Line Graph Neural Network |
| NLP | Natural Language Processing |
| PML | Probabilistic Machine Learning |
| SA | Semantic Analysis |
| SAR | Socially Assistive Robotics |
| SVM | Support Vector Machine |
| UML | Unsupervised Machine Learning |

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
