# Peer review of "Systematic Review of Computer Vision Semantic Analysis in Socially Assistive Robotics"

_ai, doi:10.3390/ai3010014_

Round 1

Reviewer 1 Report

  1. The positioning of this study is unclear, and it is not a formal review study
  2. This article does not see the relevance of section 1 and 2

Author Response

Dear,

We would like to thank the reviewers and the associate editor for their comments and criticisms, which helped us to improve the quality of our manuscript. Hoping we have brought answers to all the criticism and questions.

Please, you find in the following, our answers (in blue) to the reviewers’ comments and questions. 

Reviewer 1

  1. The positioning of this study is unclear, and it is not a formal review study

Dear Reviewer #1, thank you for questioning the clarity of the review. We have followed a specific protocol (see references [19, 20] in the paper) that has been widely deployed for review studies. We added the following paragraph in Section 1, starting at line 70, to make it clear and better position our work:

“This step towards VSA can accelerate the emergence of robotic assistants in domestic elder care, allowing for domestic burdens to be removed from the daily tasks of individuals who might not be capable of keeping up to those. Moreover, such robotic agents would guarantee a safer environment for special needs dependent users by repositioning missing, potentially dangerous items, locating lost items, or other simple but important tasks that would benefit from contextual information.”

We also better summarized our contributions in Section 5, with the following bullet-points:

“We presented the following main contributions:

  • Provide a concrete definition of Computer Vision Semantic Analysis that can be used to identify works on the fields of computer vision and semantic analysis, as described in Section 2;
  • Provide a systematic review of the current scope, research gaps, and future trends on the field of Computer Vision Semantic Analysis and its applications towards Assistive Robotics;
  • Define a novel categorization of methods in Computer Vision Semantic Analysis into 1) ontologies; 2) information merging, and 3) semantic data extraction.

  1. This article does not see the relevance of section 1 and 2

We thank the reviewer for pointing out where we could improve the paper's clarity.

Section 1 is written to explain the review of the selected intersection of research fields; we expect it to pass to our readers why VSA and its applications in assistive robotics are important.

Section 2 presents the applied protocol. We aim to show there was a scientific method strictly followed and for the review to be replicable. To further provide value to Section 2, the following new paragraphs deepen the context into our Research Questions in Section 2.2:

"To discuss the gaps and trends of Visual Semantic Analysis and its application in Assistive Robotics, we first need to understand the scope and state of such fields, search for what research and methods are currently existent in the academia, with that we defined RQs 1 and 2. 

Given that works could be found, we wondered if there are patterns of semantic data used in works combining computer vision and semantic analysis in the assistive robotics context: is there semantic information that shows itself as most valuable? Are there already defined categorizations of methods for such tasks? This questioning was designed into RQ3. 

Our third concern regards the state of lightweight visual semantic analysis models. We find it relevant to gather knowledge on whether this is a current concern of academia. Besides, we would like to analyze what metrics and techniques are used to compare and allow assistant robots to execute those visual tasks. Understanding that the current state and metrics of lightweight models in visual semantics for robotic assistance is one point-of-view and the search for methods for creating light-weight methods is another, we defined RQs 4 and 5."

Reviewer 2 Report

The authors presented "Review of Computer Vision Semantic Analysis in Socially Assistive Robotics". The paper is well written, following are a few suggestions regarding the paper:

1) The Abstract is not clear with the motivation/purpose of the review

2) Abstract is also missing with the conclusive outcomes of the paper

3) The term Compact model is not a clear (scientific term), I would suggest elaborating it further or changing it to another one

4)  Author need to provide further elaborations of the research questions RQ1-RQ5 in Section 2.2.

5) In you are using the "Compact models" term in the research question, you need to elaborate what a compact model is (is the shallow network? in terms of Layers? in terms of FLOPS? In terms of design complexity ?). And similar to the term light-weight

6) The short paragraphs should be combined to make a better view of the whole paper.

7) Add the key points of the study to the conclusion.

Author Response

Reviewer 2

The authors presented "Review of Computer Vision Semantic Analysis in Socially Assistive Robotics". The paper is well written, following are a few suggestions regarding the paper:

Dear Reviewer 2,

Thanks for the suggestions. We entirely agree, and in the following, we describe our modification to fit the suggestions, comment by comment.

1) The Abstract is not clear with the motivation/purpose of the review

We modified the abstract to include our motivation better. The following sentences were included:

“Solving those issues can provide more comfortable and safer environments for the individuals in most need. In this work, we aim to understand the current scope of science in the merging fields of computer vision and semantic analysis in the context of light-weight models for robotic assistance.”

2) Abstract is also missing with the conclusive outcomes of the paper

We also modified the abstract to include our conclusions better. The text below was added:

“We have observed that the current methods regarding Visual Semantic Analysis pave two main trends. At first, there is an abstraction of contextual data to enable automated understanding of tasks. We also observed a clearer formalization of model compaction metrics.”

3) The term Compact model is not a clear (scientific term), I would suggest elaborating it further or changing it to another one

We changed the mentions concerning “compact models” in the text to “light-weight models”. The latter truly is a more accepted term for what is discussed in the paper.

4)  Authors need to provide further elaborations of the research questions RQ1-RQ5 in Section 2.2.

More background information on each of the research questions, including motivations for each, was added to Section 2.2 through the following three paragraphs:

“To discuss the gaps and trends of Visual Semantic Analysis and its application in Assistive Robotics, we first need to understand the scope and state of such fields, search for what research and methods are currently existent in the academia, with that we defined RQs 1 and 2. 

Given that works could be found, we wondered if there are patterns of semantic data used in works combining computer vision and semantic analysis in the assistive robotics context: is there semantic information that shows itself as most valuable? Are there already defined categorizations of methods for such tasks? This questioning was designed into RQ3. 

Our third concern regards the state of lightweight visual semantic analysis models. We find it relevant to gather knowledge on whether this is a current concern of academia. Besides, we would like to analyze what metrics and techniques are used to compare and allow assistant robots to execute those visual tasks. Understanding that the current state and metrics of lightweight models in visual semantics for robotic assistance is one point-of-view and the search for methods for creating light-weight methods is another, we defined RQs 4 and 5.”

5) In you are using the "Compact models" term in the research question, you need to elaborate what a compact model is (is the shallow network? in terms of Layers? in terms of FLOPS? In terms of design complexity ?). And similar to the term light-weight

The term "Compact models" was replaced by the more commonly accepted term" light-weight models." We define light-weight models as follows: "Light-weight models can be defined as models that present a low requirement for hardware capabilities, displayed in specific metrics such as FLOPs, number of parameters, storage size, and memory usage.". We have added this information to the text. We based this definition on metrics presented in previous works aimed towards mobility, such as [53].

6) The short paragraphs should be combined to make a better view of the whole paper.

Sequences of paragraphs shorter than four lines were combined when not disruptive to the reading.

7) Add the key points of the study to the conclusion.

The following bullet-point list of main contributions was added to Section 5:

“We presented the following main contributions:

  • Provide a concrete definition of Computer Vision Semantic Analysis that can be used to identify works on the fields of computer vision and semantic analysis, as described in Section 2;
  • Provide a systematic review of the current scope, research gaps, and future trends on the field of Computer Vision Semantic Analysis and its applications towards Assistive Robotics;
  • Define a novel categorization of methods in Computer Vision Semantic Analysis into 1) ontologies; 2) information merging, and 3) semantic data extraction.
